# Identifying post-marital residence patterns in prehistory: A phylogenetic comparative analysis of dwelling size

Václav Hrnčíř[1,2]☯*, Pavel Duda[3]☯, Gabriel Šaffa[3], Petr Květina[2], Jan Zrzavý[3]

**1** Department of Archaeology, Faculty of Arts, Charles University, Prague, Czechia, **2** Institute of Archaeology of the Czech Academy of Sciences, Prague, Czechia, **3** Department of Zoology, Faculty of Science, University of South Bohemia, České Budějovice, Czechia

☯ These authors contributed equally to this work.
* hrncir.vaclav@gmail.com

**Data Availability Statement:** All relevant data are within the manuscript and its Supporting Information files.

## Abstract

Post-marital residence patterns are an important aspect of human social organization. However, identifying such patterns in prehistoric societies is challenging since they leave almost no direct traces in archaeological records. Cross-cultural researchers have attempted to identify correlates of post-marital residence through the statistical analysis of ethnographic data. Several studies have demonstrated that, in agricultural societies, large dwellings (over ca. 65 m$^2$) are associated with matrilocality (spouse resides with or near the wife's family), whereas smaller dwellings are associated with patrilocality (spouse resides with or near the husband's family). In the present study, we tested the association between post-marital residence and dwelling size (average house floor area) using phylogenetic comparative methods and a global sample of 86 pre-industrial societies, 22 of which were matrilocal. Our analysis included the presence of agriculture, sedentism, and durability of house construction material as additional explanatory variables. The results confirm a strong association between matrilocality and dwelling size, although very large dwellings (over ca. 200 m$^2$) were found to be associated with all types of post-marital residence. The best model combined dwelling size, post-marital residence pattern, and sedentism, the latter being the single best predictor of house size. The effect of agriculture on dwelling size becomes insignificant once the fixity of settlement is taken into account. Our results indicate that post-marital residence and house size evolve in a correlated fashion, namely that matrilocality is a predictable response to an increase in dwelling size. As such, we suggest that reliable inferences about the social organization of prehistoric societies can be made from archaeological records.

## Introduction

Post-marital residence rules specify where a person resides after marriage and, accordingly, influence social organization of human societies. In modern wage-based economies, most

**Funding:** V.H, P.D., G.Š. and J.Z. were supported by the Czech Science Foundation (GACR) Grant 18-23889S. P.K. was supported by the Czech Science Foundation (GACR) Grant 19-16304S. G.Š. was supported by GAJU project 048/2019/P. The funders had no role in study design, data collection and analysis, decision to publish, or preparation of the manuscript.

**Competing interests:** The authors have declared that no competing interests exist.

newlyweds tend to establish a new household separate from their respective families (neolocal residence). However, in traditional societies couples typically live with or near one's parents [1]. About 71% of all societies listed in the *Ethnographic Atlas* [2] are predominantly patrilocal, while 11% are matrilocal. Ambilocality, multilocality, avunculocality, and neolocality are less frequent, together accounting for the remaining 18% of societies [3]. However, the same distribution does not apply to hunter-gatherer societies, which have a more flexible social organization, being most frequently ambi-/multilocal [4]. The decision regarding who will leave home after marriage and who will stay with their own kin affects many important aspects of social organization [1], including descent systems and kinship terminology [5], wealth inheritance rules [6], modes of marriage [3], community size [7], division of labor [8], migration [9], and warfare [10, 11]. Murdock argued that "when any social system undergoes change, such change regularly begins with a modification in the rule of residence" [5, p. 221]. This notion that a change of post-marital residence rule drives change in other aspects of social organization, not *vice versa*, has become known as "Main Sequence Theory".

Exact definitions of post-marital residence patterns vary considerably [12]. Some scholars use "patrilocality" as a general term for residence with or near the husband's family [e.g. ref. 1, 13], while others [e.g. ref. 14, 15] distinguish between "patrilocality" (residing in the husband's father's household) and "virilocality" (residing with the husband's kin in a more general sense). The same applies to "matrilocality" and "uxorilocality", referring to residence with or near the wife's kin. "Ambilocality" refers to residence with or near the kin of either spouse, while "multilocality" refers to the situation where couples move between the households of both sets of parents. "Avunculocality" can be considered a special case of virilocality, when a couple lives with the husband's maternal uncle. In this study, we use "patrilocality" for both patri- and virilocality and "matrilocality" for both matri- and uxorilocality, since they cannot be distinguished in prehistoric patterns by the currently available methods.

Scholars have employed various methods of identifying post-marital residence patterns in prehistoric societies from archaeological records. Primarily, they have focused on skeletal morphology, since inferring any kind of social organization from the variability or spatial distribution of material culture can be misleading [16, 17]. Traditionally, bioarchaeologists examined morphological variation in skeletal and dental traits to identify differences between males and females [for an extensive review of this approach see ref. 14]. According to the theory, the sex with the greater within-group morphological variability is assumed to be the more mobile one. For instance, greater female variability corresponds to greater female migration and thus could indicate patrilocality.

With more recent advances in scientific methods, the focus of bioarchaeologists has moved to isotopic and ancient DNA analyses. For example, researchers using strontium isotope analysis of human tooth enamel [18] found significantly more variance in the distribution of $^{87}Sr/^{86}Sr$ signatures among females than among males in early Neolithic Central Europe (5500–5000 BC), indicating patrilocality during this historical period [19]. The same residence pattern has also been proposed for the late Neolithic (2700–2400 BC) communities in Eulau [20], Bergrheinfeld and Lauda-Königshofen [21], and Early Bronze Age (2150–1700 BC) Lech River valley [22], all in Germany, where females fall outside the local strontium range, indicating that their place of birth (and childhood) was elsewhere. Similarly, but in the opposite direction, isotopic evidence suggests a possible transition to matrilocality during the second millennium BC in Thailand [23, 24]. Sex-biased mobility differences can also be inferred from ancient DNA sequences (specifically mtDNA and Y-chromosomal haplotypes). Patrilocal societies should have relatively lower Y-chromosomal diversity and larger mtDNA diversity within a population, while the opposite pattern is expected for matrilocal societies. This has been demonstrated in present-day patrilocal and matrilocal groups in northern Thailand [25] and

also applied in archaeology, e.g. for suggesting that Neanderthals in Iberia (ca. 49,000 BP) were patrilocal [26], or that the prehistoric North American Hopewell community (100 BC to AD 400) was matrilocal [27]. However, more recent studies have shown that the association between DNA diversity and post-marital residence pattern is much less straightforward and not universal [28–31]. Other attempts to infer past social organization are based on population genetic analyses. For example, an abrupt reduction in Y-chromosomal diversity (compared to mtDNA) inferred across several Old World populations around 8,000–4,000 BP [32], has been interpreted as evidence of predominant patrilineality and patrilocality during this period [33].

Anthropologists have applied phylogenetic comparative methods, adopted from evolutionary biology [34, 35], to reconstruct the evolution of cultural traits. Using language trees as a proxy for historical relationships between populations, the evolution of post-marital residence rules has been reconstructed in Austronesian [13, 36], Bantu [37], Indo-European [15, 36], and Tupi [38] language families. The results of these studies suggest that early Austronesians were matrilocal and matrilineal, the first Bantu were patrilocal and patrilineal, early Indo-Europeans practiced patrilocality and/or neolocality, and Tupi ancestors were matrilocal. Recently, Moravec et al. [39] modelled transitions in post-marital residence rules in five language families (Austronesian, Bantu, Indo-European, Pama-Nyungan, and Uto-Aztecan) and found that there is no universal pattern of evolution for post-marital residence rules, although patrilocality seems to be the most common state across space and time. Apart from reconstructing the history of various cultural practices, the phylogenetic comparative approach is useful for studying associations between cultural traits, while controlling for phylogeny. For example, Jordan [40] demonstrated that in Austronesian societies, changes in post-marital residence preceded changes in descent systems, whereas Opie et al. [37] found that in Bantu societies, a change in descent system was always followed by a shift away from the ancestral post-marital residence state. Surowiec et al. [41] found, using a worldwide sample of societies, that matrilineal descent emerges first, followed by a shift towards matrilocality, more often than *vice versa*, challenging Murdock's [5] Main Sequence Theory. Walker et al. [42] demonstrated that the prevalent belief in partible paternity is associated with matrilocal residence in Carib, Macro-Je, Pano, and Tupi language families.

Cross-cultural researchers have attempted to identify correlates of post-marital residence patterns through statistical analysis of ethnographic data [43]. The association between average house floor area (AHFA) and post-marital residence (PMR) was first demonstrated by Ember [44]. In his seminal paper, he showed, using two cross-cultural samples, that AHFA in matrilocal societies is usually more than 51–56 m$^2$, while the majority of patrilocal societies have smaller houses. (Note that we use "house" and "dwelling" interchangeably in this paper, both terms referring to residential building). Subsequent studies by Divale [45] and Brown [46] confirmed his findings. According to Divale [45], any archaeological site that had an AHFA less than 42.7 m$^2$ could be inferred to have had patrilocal residence with 95% confidence. Conversely, an AHFA larger than 79.2 m$^2$ indicates a matrilocal residence. Brown [46] did not suggest any cut-off value; nevertheless, his test confirmed the correlation. Mean AHFA values in his sample were 27.4 m$^2$ for patrilocal societies and 78.4 m$^2$ for matrilocal ones. Two decades later, Porčić [47] tested these findings. He combined all data from the previous studies into a larger sample of 80 societies and added a new variable into the analysis: the mode of subsistence. His results confirmed the association between AHFA and PMR, but the mode of subsistence had a significant effect on the correlation. The AHFA-PMR association was only significant in agricultural societies, improving the prediction rate by almost 25%, but not in foraging or pastoral societies. This finding was positively received by archaeologists, as dwelling size is usually easy to determine, and has been applied to various archaeological contexts, e.g. to historical northern Iroquoian groups (AD 500–1300; [48]), Chaco Canyon region (AD

900–1150; [49–52]), Hohokam culture (AD 0–1450; [53, 54]) and Neolithic Greece (6600/ 6500–3300 BC; [55]).

In order to explain his findings, Ember [44] argued that matrilocal societies tended to have larger houses because married sisters find it easier to live together than non-sisters and thus these societies tend to form larger households. In Divale's [45] opinion, larger matrilocal households enhance trust and cooperation between unrelated brothers-in-law who did not know each other before marrying into the community. In this respect, large matrilocal households serve a similar function as men's houses, where men from different families eat, work and sleep together. According to Porčić [47], the absence of agriculture generally implies more mobile subsistence strategies such as foraging or pastoralism. People in mobile societies tend to spend less time and energy building houses and thus have smaller dwellings made of lighter materials, regardless of their post-marital residence rules.

However, these studies suffer from several methodological issues. First, they only considered two types of PMR: matrilocal and patrilocal. Neolocality was excluded from Ember's original study because he found that it correlated with the presence of monetary exchange and markets [56]. Ambilocality and multilocality were also not considered because another cross-cultural study [57] found them to be associated with recent depopulation. Avunculocality was omitted simply because it is rare, present in less than five percent of world cultures [44].

Second, previous studies did not control for the non-independence of societies due to common ancestry. As Galton pointed out in the 19[th] century [see the discussion in ref. 58], societies cannot be treated as statistically independent. Similar cultural traits can reflect convergent adaptations to similar socio-ecological pressures as well as common ancestry. This realization later became known as "Galton's problem". Anthropologists have attempted to minimize Galton's problem by using subsets of distantly related societies that were assumed to be effectively independent, such as the Standard Cross-Cultural Sample [59]. However, failure to take relatedness into account leads to elevated Type I and Type II error rates, even in the datasets designed for the purpose of mitigating Galton's problem [60, 61]. Common ancestry can be accounted for with a use of phylogeny, which captures the expected covariance among societies. It allows not only to test for a correlation between AHFA and PMR while controlling for non-independence, but also to detect independent (convergent) changes in AHFA in response to changes in PMR or other aspects of social organization. Using phylogenetic comparative methods, we can determine whether large houses are a predictable response to matrilocality and whether AHFA can inform us about the social organization of prehistoric societies.

Moreover, except the presence of agriculture in Porčić's study [47], other aspects that could impact AHFA were not considered. Although Porčić also assumed that house construction material and settlement patterns can significantly affect the house size, he did not include these variables into his analyses. Household wealth is another factor which is positively correlated with house size in many societies [for references see 62, S1 File]), indicating that large dwelling does not always mean more household members. Apart from residential and symbolic functions, exceptionally large dwellings could also serve other purposes, such as storage, meeting, defensive or ritual. The appearance and size of dwellings could be significantly influenced also by sociopolitical settings and colonialism. Some types of building materials and technologies could have made it possible to build larger houses, while intercultural contact could have led to change of architectural style.

In the present study, we re-examine the association between the AHFA and PMR using a different sample of societies, revised AHFA values, and a finer continuous variable that captures all types of PMR. Our analysis includes additional explanatory variables, specifically the presence of agriculture, fixity of settlement, and house construction material, while controlling

for non-independence using a time-calibrated phylogenetic supertree of human populations based on genetic and linguistic data [63, 64].

## Methods

### Study variables

The AHFA data for 80 societies were taken from Porčić's study [47], which were collected from three previous studies [44–46]. We added 22 new populations for which the AHFA was reported by Brown [46] but not included in previous analyses because they were not (strictly) patrilocal or matrilocal (see S1 File). Where possible, we checked the data against their original sources (see S1 File). AHFA values were log-transformed to ensure a normal-like distribution of the data.

Data on post-marital residence rules, the presence of agriculture, fixity of settlement, and construction material were obtained from the open-access *Database of Places, Language, Culture, and Environment* (*D-PLACE*; [65]). All study variables are described in Table 1 (see also Table A in S1 File). The variable "Marital residence with kin: prevailing pattern [EA012]" was chosen as a proxy for post-marital residence because the same variable in the *Ethnographic Atlas* [2] was used in previous studies and it is more finely-resolved than the similar variable "Transfer of residence at marriage: prevailing pattern [EA011]". Original categories were reduced to a five-point scale, which captures a tendency towards matrilocality.

**Table 1. Description of study variables.**

| Name | Original source | Original scale | Transformation |
|---|---|---|---|
| AHFA (ord) | Ref. [47] or primary sources in S1 File | Continuous measure between 0 and ∞ | Log-transformed to ensure a normal-like distribution of the data |
| AHFA (bin) | "As above" | "As above" | Dichotomized into small ($< 65$ m$^2$) and large ($> 65$ m$^2$) |
| PMR (ord) | *D-PLACE*–Marital residence with kin: prevailing pattern [EA012] | 1 = Avunculocal | Reduced to five-state continuous trait indicating tendency towards matrilocality: 0 = 1, 4, 8, 10 on original scale |
| | | 2 = Ambilocal | 1 = 12 |
| | | 3 = Avuncu-uxorilocal | 2 = 2, 3, 6, 7 |
| | | 4 = Avuncu-virilocal | 3 = 11 |
| | | 5 = Matrilocal | 4 = 5, 9 |
| | | 6 = Neolocal | |
| | | 7 = Separate | |
| | | 8 = Patrilocal | |
| | | 9 = Uxorilocal | |
| | | 10 = Virilocal | |
| | | 11 = Ambi-uxo | |
| | | 12 = Ambi-viri | |
| PMR (bin) | "As above" | "As above" | Dichotomized into non-matrilocal (1–4, 6–8, 10, 12 on original scale) and matrilocal (5, 9, 11) |
| Agriculture | *D-PLACE*–Agriculture: intensity [EA028] | 1 = No agriculture | Dichotomized into agriculture not important (1–2 on original scale) and agriculture important (3–6) |
| | | 2 = Casual agriculture | |
| | | 3 = Extensive or shifting agriculture | |
| | | 4 = Horticulture | |
| | | 5 = Intensive agriculture | |
| | | 6 = Intensive irrigated agriculture | |

*(Continued)*

**Table 1.** (Continued)

| Name | Original source | Original scale | Transformation |
|---|---|---|---|
| Settlement | *D-PLACE*–Settlement patterns [EA030] | 1 = Nomadic bands | Dichotomized into mobile (1–2 on original scale) and sedentary (3–8) |
| | | 2 = Seminomadic communities | |
| | | 3 = Semisedentary communities | |
| | | 4 = Impermanent settlement | |
| | | 5 = Dispersed homesteads | |
| | | 6 = Hamlets | |
| | | 7 = Villages/towns | |
| | | 8 = Complex settlements | |
| Material | *D-PLACE*–House construction: wall material [EA081] or House construction: roofing materials [EA083][a] | 1 = Stone, stucco or brick | Dichotomized into impermanent material (2,4,5,6,7,8,10 on original scale and 10 from variable EA083) and durable material (1,3,9 and 9 from variable EA083) |
| | | 2 = Plaster, clay or similar | |
| | | 3 = Wood or bamboo | |
| | | 4 = Bark | |
| | | 5 = Hides or skins | |
| | | 6 = Fabric | |
| | | 7 = Mats | |
| | | 8 = Grass | |
| | | 9 = Adobe, clay or brick | |
| | | 10 = Open walls | |
| | | 9[EA083] = Earth or turf | |
| | | 10[EA083] = Ice or snow | |

[a]Populations with a character state 11 = "walls indistinguishable from roof or merging into the latter" in variable [EA081] were scored based on variable [EA083].

Three additional explanatory variables were dichotomized: agriculture into "agriculture not important" and "agriculture important" according to Porčić [47]; settlement into "mobile" and "sedentary" indicating fixity of settlement; and material into "impermanent" and "durable" indicating durability of wall material of the prevailing type of dwelling.

AHFA and PMR were additionally dichotomized in order to test for correlated evolution (see Phylogenetic comparative analysis). AHFA was coded as "small" and "large", with a cut-off value of 65 m$^2$ as per Porčić [47]; PMR was coded as "non-matrilocal" and "matrilocal".

## Phylogenetic comparative analysis

To apply phylogenetic methods to our global sample of societies, we leveraged a time-calibrated supertree of human populations [63, 64]. This supertree (i.e. a tree of trees) was based on 388 genetic and linguistic phylogenies published between 1990 and 2017, and time-calibrated using 265 node-age constraints derived from genetic, linguistic, archaeological, historical, and epigraphic data. A subset tree of 86 populations (from a total of 102; the others were not included in the phylogeny) for which AHFA values were available was used as a phylogenetic control (Fig 1, Table A in S1 File). We measured phylogenetic signal of individual continuous and binary traits using Pagel's λ [66] and Fritz and Purvis's D [67], respectively. The λ values for each multistate trait were estimated using the *phylosig* function in the R package *phytools* [68]. D values for each trait in our sample were estimated using the *phylo.d* function in the R package *caper* [69]. The maximum likelihood (ML) reconstruction of ancestral states was

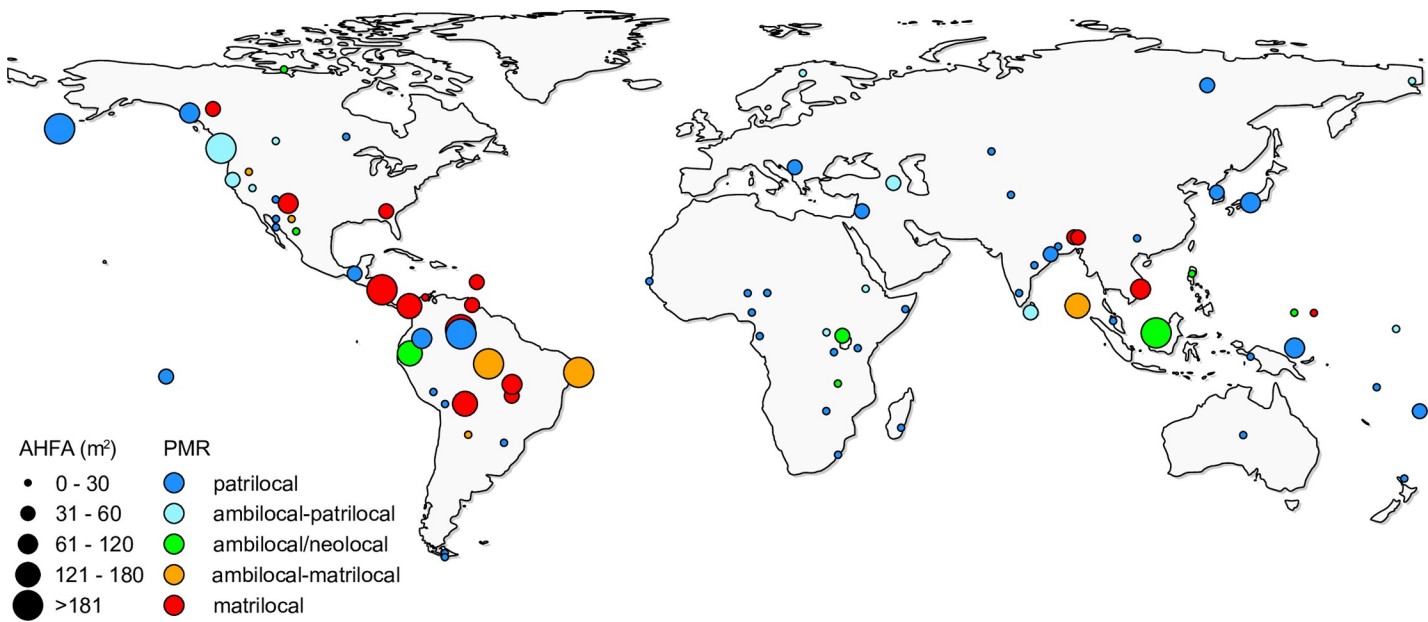

**Fig 1. World map showing the distribution of the 86 sample societies.** Dot size corresponds to the average house floor area (AHFA); colors indicate the post-marital residence (PMR) pattern.

performed using the *fastAnc* function and the resulting estimates were plotted using the *contMap* function in the R package *phytools* [68].

We modelled a probability that AHFA is a linear function of the explanatory variables, using phylogenetic generalized least squares (PGLS) regression as implemented in the *pgls* function of the R package *caper* [69], while simultaneously controlling for phylogenetic signal (as measured by the ML estimate of λ) in the residuals of each model. We assessed the explained variance by the model with an adjusted coefficient of determination ($R^2$) and based our model selection on the Akaike information criterion (AIC).

We tested for correlated evolution between dichotomized (binary) versions of AHFA and PMR using Pagel's [70] test for correlated evolution as implemented in the *fitPagel* function of the R package *phytools* [68]. Pagel's method assumes a correlation between two binary traits when the dependent, eight-parameter model, in which the probability of change in one trait depends on the state of the other trait, fits the data better that the independent, four-parameter model, in which evolution in each character is independent of the state of the other character. A goodness-of-fit test based on a likelihood ratio was used to compare log likelihoods of the two models.

## Results

All independent variables showed a relatively low but non-random phylogenetic signal (Table 2). The dependent variable AHFA displayed an effectively random phylogenetic structure (λ = 0.103, p = 0.269).

The ML reconstruction of ancestral states (Fig 2) indicates that the last common ancestor of sample societies had very small houses (11.7 m²) and was patrilocal (point estimate 0.3 on a scale from 0 to 5). There is a general tendency towards an increase in AHFA. Dwelling size has decreased in only a few lineages (e.g. aboriginal Australians, populations of Patagonia and Tierra del Fuego, and Maori people in New Zealand; Fig 2A). The reconstruction indicates multiple independent increases in AHFA in societies that shifted towards a more flexible

**Table 2. Phylogenetic signal of study variables.**

| Variable | Phylogenetic signal | p-value |
|---|---|---|
| AHFA (ord) | λ = 0.103 | 0.269 |
| PMR (ord) | λ = 0.139 | 0.031* |
| Agriculture | D = 0.383 | < 0.001* |
| Settlement | D = 0.732 | 0.037* |
| Material | D = 0.767 | 0.037* |

Pagel's λ for continuous variables (λ values are between 0 and 1, where 0 indicates no phylogenetic signal) and Fritz and Purvis's D for binary variables (D values are also between 0 and 1, but with 1 indicating no phylogenetic signal.); p ≤ 0.05 indicates that we can reject the "random distribution" hypothesis.

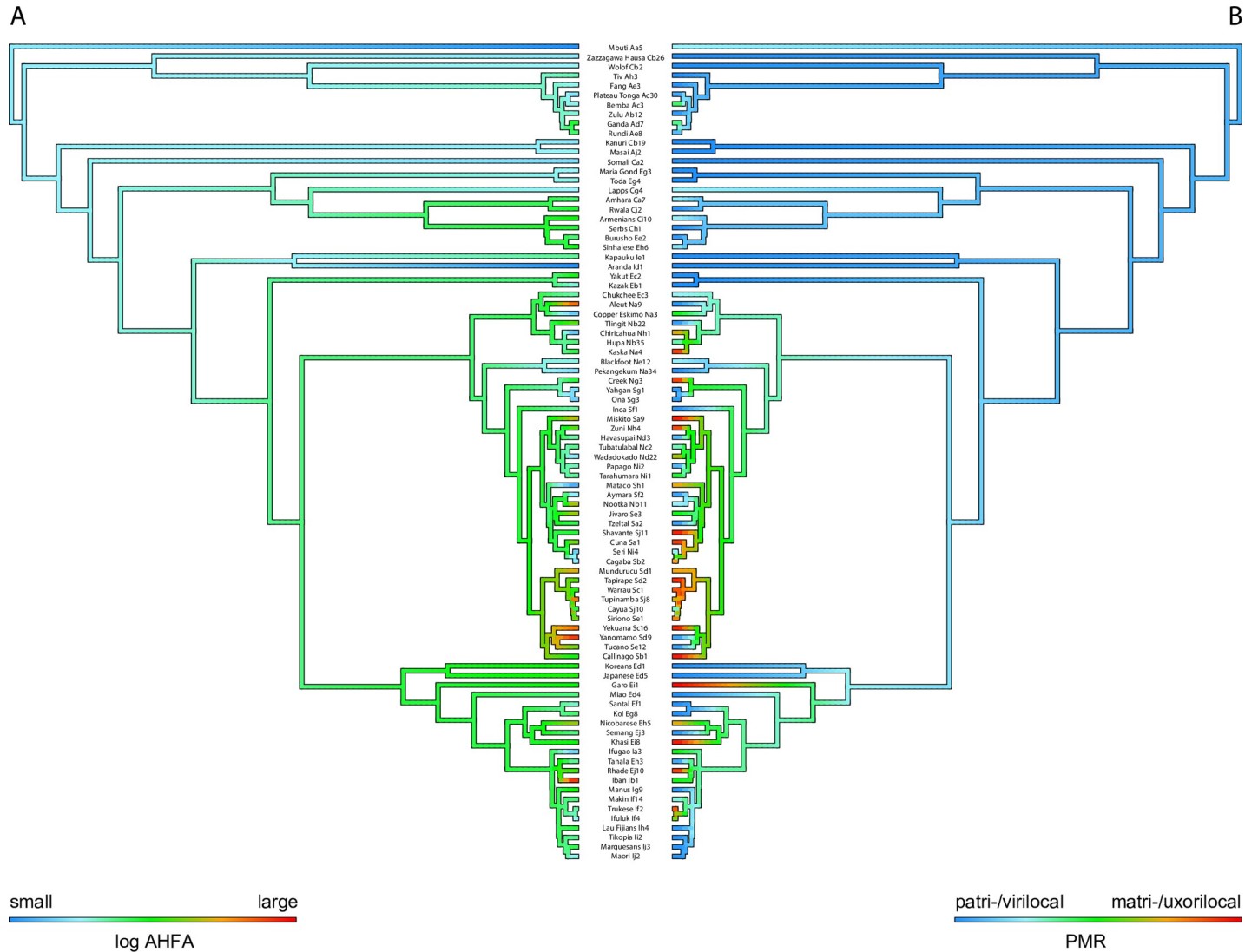

**Fig 2.** The evolution of (A) AHFA and (B) PMR across the phylogeny. Colors of internal branches correspond to the inferred ancestral state based on maximum likelihood reconstruction of ancestral states in the R package *phytools*.

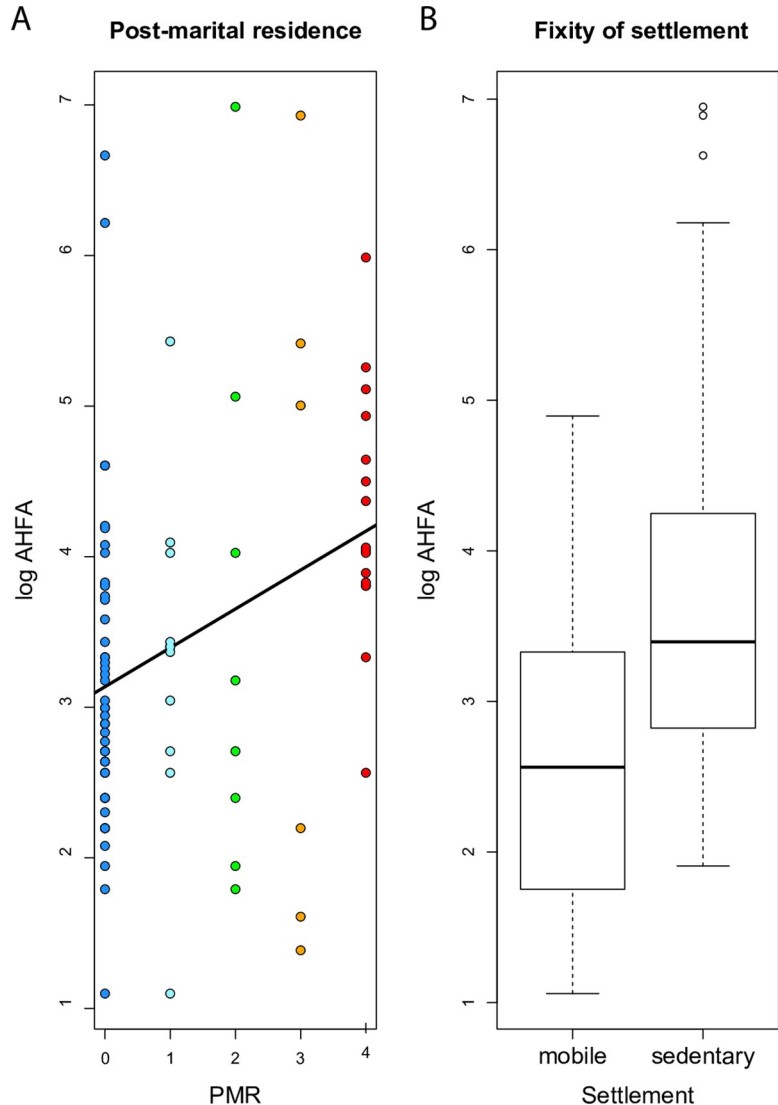

**Fig 3.** The association between (A) AHFA and PMR and (B) AHFA and settlement. The color coding for PMR states corresponds to Fig 1.

(ambi-/neolocal) post-marital residence pattern and towards matrilocality in North and South America and in East Asia (Fig 2B).

AHFA and PMR are significantly positively associated according to the PGLS analysis (Fig 3A, Table 3). The same result is obtained when a binary version of PMR is used, which is more comparable to the previous study by Porčić [47]; however, both p and adjusted $R^2$ values are lower. The five-state continuous trait explains about 10% of the total variance in AHFA. AHFA also shows a positive association with agriculture, but only when the binary version of the trait is used (Table 3). The association between AHFA and agriculture loses significance once fixity of settlement is taken into account. The settlement is the single best predictor of AHFA (Fig 3B, Table 3), explaining about 16% of the total variance. The single best model (with highest $R^2$ and lowest AIC) is the one that combines AHFA and PMR with settlement (Table 3). Construction material is not significantly associated with AHFA.

**Table 3. Model comparison for AHFA.** Models include different explanatory variables, differently coded variables, different combination of variables, and phylogenetic control.

| Model | p-value (F-statistic) | Adjusted $R^2$ | AIC |
|---|---|---|---|
| AHFA~PMR (ord) | 0.002* | 0.097 | 275.8615 |
| AHFA~PMR (bin) | 0.004* | 0.086 | 276.9319 |
| AHFA~Agriculture (bin) | 0.003* | 0.085 | 275.8165 |
| AHFA~Agriculture (ord) | 0.328 | 0.000 | 283.4568 |
| AHFA~Settlement | < 0.001* | 0.163 | 268.1564 |
| AHFA~Material | 0.091 | 0.022 | 281.5124 |
| AHFA~PMR (ord) + Settlement | < 0.001* | 0.235 | 261.7224 |
| AHFA~PMR (ord) + Settlement + Agriculture (bin) | < 0.001* | 0.235 | 263.4807 |
| AHFA~PMR (ord) + Settlement + Material | < 0.001* | 0.239 | 263.0849 |
| AHFA~PMR (ord) + Agriculture (bin) + Settlement + Material | < 0.001* | 0.231 | 264.8841 |

The test for correlated evolution indicates that AHFA and PMR are indeed correlated on phylogeny (p < 0.001). The model with PMR as a dependent variable provides the best fit to the data (Fig 4, S1 Table), indicating that the change in house size precedes the change in residence. The combination of small house and patrilocal residence is both the ancestral state and evolutionarily the most stable state. The combination of large houses and patrilocal residence as well as small houses with matrilocal residence are evolutionarily unstable, resulting in a change of house size or a change of post-marital residence rule. It is rare for a matrilocal society with large houses to transition directly to patrilocality; decreases of house size are more common in matrilocal societies and these are generally followed by the transition to patrilocal residence (Fig 4).

## Discussion

### Cross-cultural association between matrilocality and house size

Post-marital residence is not an isolated aspect of human social organization but is closely tied to other social structures. Societies with larger houses tend to be matrilocal (although very

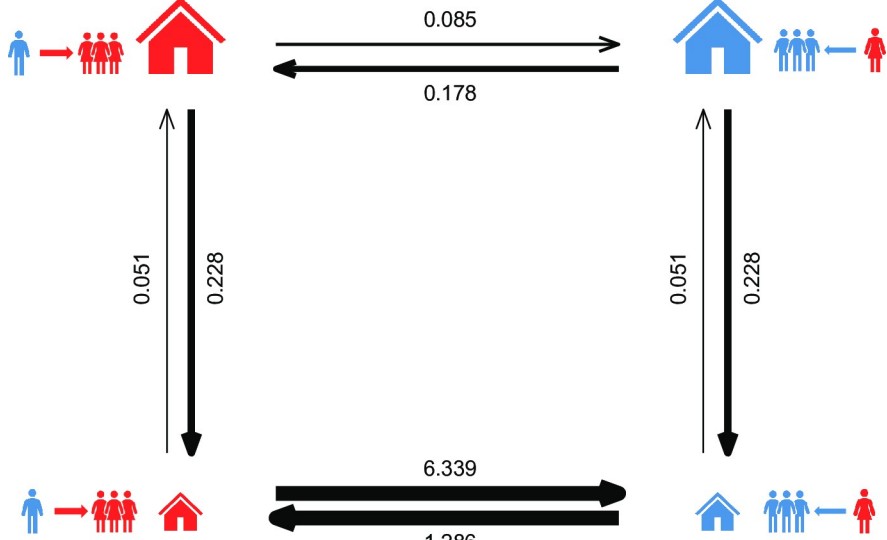

**Fig 4. Transition rate matrix for the correlated evolution between AHFA (dependent variable) and PMR.** Widths of arrows are proportional to rates of change.

large dwellings can be associated with any type of residence pattern). The association remains significant even when the historical relatedness of sampled societies is controlled for and multiple explanatory variables are included in the model.

In contrast to previous studies [44–47], we applied phylogenetic comparative methods. The previous sample compiled by Porčić [47] was geographically imbalanced, consisting mainly of closely related American societies that shared a common ancestor no more than 16,000 years ago [71]. Our results confirm that these societies are indeed not statistically independent. In our study, all variables except AHFA showed a non-random (although relatively low) phylogenetic signal.

In our sample, although AHFA had a globally random phylogenetic structure, a detailed view at the local level showed that closely related populations often built similar dwellings. For example, there were several regional architectonic traditions of large houses in North America. Longhouses were typical for Iroquoian cultural groups [72], circular earth-covered lodges were known from tribes of the Plains [73], and hardwood plankhouses could be found among hunter-gatherers on the Northwest Coast [74]. Three studied South American Tupi-Guaraní populations, namely Mundurucu [75], Tapirape [76], and Tupinamba [77], also lived in similar dwellings: large rectangular houses with walls made of bark or palm leaves, arranged around a central village plaza.

The single best predictor of AHFA is the fixity of settlement (Table 3, Fig 3B); mobile populations prefer to live in small, easy to build houses. Agriculture, when coded as a binary trait ("not important" or "important"), was found to have a positive association with AHFA, as has been previously documented [47], but this association loses significance once the fixity of settlement is included into the model. Although it is true that "the presence or absence of agriculture should be less difficult to infer archaeologically than mobility patterns" [47, p. 408], both traits are not in perfect correlation. While the majority of pastoralists and hunter-gatherers are indeed quite mobile, foragers subsisting predominantly on fishing are more sedentary [78], such as those of the northwest coast of North America living partially or fully sedentarily in large houses [79].

Construction material was not found to be significantly associated with AHFA. This can be partially explained by a less than ideal choice of variable to represent durability of house construction material in our study. For example, in houses with framed constructions, framing material is a much better indicator than wall material (e.g. long houses of Tupinamba, which were occupied for several years, were made of palm thatch on a wooden frame [77]). Unfortunately, framing material is not coded in *Ethnographic Atlas* [2] or *D-PLACE* [65]. Nevertheless, the architectural tradition of large houses might be influenced by the availability and quality of building materials. For example, "ironwood" was essential for longhouses of Borneo [80], cedar wood for Pacific Northwest plank houses [74], and the long leaves of the motacú palm for the simple but quite large dwellings of the Siriono people [81].

Previous studies [44–47] documented the correlation between dwelling size and post-marital residence; our results are in support of their findings (Fig 3A). Large houses usually indicate large households and these might be preferentially occupied by married sisters rather than non-sisters [44]. This argument is based on the finding that in polygynous societies, sororal co-wives usually live together in the same house, while nonsororal co-wives tend to live in separate houses, or at least in separate apartments of the large dwelling [5, pp. 30–31]. Instability of households where brothers and their spouses co-reside (so-called patrilocal joint families) was also documented, for example, in India [82, p. 106] or pre-revolutionary China [83, pp. 402–403]. In both cases, co-resident nuclear families usually broke up after the death of the father; the division was often accelerated by quarrels between wives. On the other hand, even sisters are not immune to verbal and physical aggression towards each other. Cross-

cultural survey found that sisters are often aggressive towards each other in eight percent of societies, and probably in additional eight percent of societies when they are co-wives, while aggression between sisters-in-law is not substantially higher, in 14 percent of societies [84].

Another, not necessary competing, explanation is that large households improve the integration of unrelated brothers-in-law into a matrilocal community [45]. It might also be true that traditional residence typologies do not reflect the true complexities of ethnographic variation (see below). In some of the so-called matrilocal societies, men and women spend more time with their kin at different community levels; for example, while men spend more time with kin at the village level, possibly to facilitate male alliances, women spend more time with kin at the extended household level, possibly to facilitate allomaternal care [85].

Notably, societies with very large AHFA (over ca. 200 m$^2$) were not associated with any particular type of residence. There are seven such societies in our sample. Three of them are patrilocal or predominantly patrilocal (Aleut, Nootka, Yanomamo), three are matrilocal or predominantly matrilocal (Makitare, Mundurucu, Tupinamba), and one is ambilocal (Iban). All of these were sedentary populations, but with different subsistence economies. The majority of them practiced extensive or shifting agriculture, although Aleut and Nootka were hunter-fisher-gatherers. None of them kept cattle, and only the Iban kept pigs (but note the cattle-keeping Miskito with the eighth largest AHFA in the sample, just below the 200 m$^2$ boundary). Except for Iban and Aleut, all societies with very large dwellings are from North or South America. Out of the 16 societies with known AHFA that were not included in the phylogeny (see Table A in S1 File), an additional four societies, all from North America, lived in dwellings with an AHFA of over 200 m$^2$; three are matrilocal (Huron, Iroquois, Pawnee) and one is ambi-patrilocal (Bellacoola).

These examples show that the relationship between house size and post-marital residence is not straightforward and some other factors might influence household composition than those suggested above. In societies with very large houses, one household usually consisted of multiple families (e.g. up to 30 in Tupinamba [77], up to 40 in Aleut [86], or up to 50 in Iban [87]), and it can be assumed that such large units were more resistant to dissolution due to disputes between individuals, than smaller households consisting of only two or three families. In a larger household, there were more mediators and authorities who could settle a dispute. Moreover, leaving of one family did not led to disintegration of the entire household.

The best model combined AHFA, settlement and PMR (Table 3). Smaller houses are associated with a migratory lifestyle and patrilocal residence, while large houses are typical for matrilocal sedentary societies. However, it is difficult to establish causality. Does the transition to matrilocality lead to larger dwellings, or does the increase in dwelling size lead to changes in the rule of residence? Divale's [45] argumentation, i.e. that the function of large households is to enhance trust and cooperation between unrelated brothers-in-law, suggests the former possibility. Ember [44] argues that large houses are preferentially occupied by women and their kin, indicating the latter. Our global phylogenetic analysis seems to support Ember's argumentation. Pagel's test for correlated evolution, based on binary traits, indicates that the increase of dwelling size is followed by transition to matrilocality, rather than *vice versa* (Fig 4). However, these results must be interpreted with caution. The dichotomization of continuous traits comes with a loss of information. The dependence of PMR on AHFA could be partially explained by the inability to reconstruct ancestral PMR unambiguously in deeper nodes. That said, the reconstruction of ancestral states based on continuous traits also indicates that the AHFA increased before multiple independent transitions to matrilocality occurred. The reconstruction indicates that the last common ancestor lived in very small houses (ca. 12 m$^2$, close to dwelling size in African societies in our sample, such as Bemba, Fang, Masai, or Wolof). AHFA has increased steadily throughout history, regardless of social organization.

There is probably no universal explanation for the change in the dwellings size and/or in post-marital residence rules. It has been proposed that changes in post-marital residence rules can be initiated by migration [9], depopulation [57], or the emergence of commercialization [56]. A non-matrilocal residence is predicted by a very low female contribution to subsistence [8] or by internal (rather than purely external) warfare [10, 11]. Other crucial factors can include the presence of alienable property and paternity uncertainty [88]. Matrilineal and matrilocal social structures are negatively correlated with intensive agriculture [41, 89] and heritable forms of wealth (e.g. land, money, slaves or large domestic animals [5, 41, 89, 90]) in addition to lower levels of paternity confidence [91, 92]. Specifically, in lowland South American societies, matrilocality often co-occurs with belief in partible paternity, i.e. that more than one biological father can contribute to the formation of a fetus [42].

## Reconstructing post-marital residence patterns in prehistoric societies: limitations of phylogenetic cross-cultural analyses

Our results suggest that average house floor area can be used as a proxy for post-marital residence pattern in prehistoric societies. However, before we start hypothesizing about post-marital residence in particular society, we must consider the limitations of cross-cultural studies.

This study, as well as previous analyses [44–47], depend on data from ethnographic literature, which primary focus is usually not the size of dwellings or post-marital residence patterns. References to these cultural traits are often anecdotal and not resulting from empirical research. Data in large ethnographic databases (such as *D-PLACE* [65]) ordinarily capture each culture at a particular time (and location), making backward verification difficult. The sizes of dwellings can be re-examined archaeologically in some areas, but regarding post-marital residence, one must rely on the original ethnographic records. As the Goodenough-Fischer controversy on the Trukese marital residence demonstrated, ethnographers' conclusions can be sometimes contradictory, even when researchers compile a house to house censuses [16].

Using AHFA as a predictor variable is practical from an analytical perspective, but it sometimes simplifies the real situation. The range of house floor area can be wide, especially among societies with large houses, e.g. 70–900 m$^2$ among Aleut [93], 20–110 m$^2$ (exceptionally more than 900 m$^2$) among Garo [94, 95], and 100–500 m$^2$ among Tucano [96–98]. It is usually the case that no data are available on differences in household composition between the smallest and largest households in these societies, and it is not clear whether house size can affect post-marital residence within a population. It is also important to consider how much the size of a house reflects the size of a household. For example, a residential building does not necessarily represent a single space, whether in functional or social contexts. It can be divided into several apartments (e.g. in Iban longhouses [87]) or it can include non-residential parts (e.g. stables in German hall houses [99]). Some residential dwellings (e.g. those belonging to community leaders) can serve multiple functions, for example, as a storage area or as a venue for council meetings, feasts, ceremonies and other social gatherings. Furthermore, the house does not need to be inhabited by a nuclear or extended family members only. For example, among the Mundurucu, all post-pubescent men, single and married, relaxed and slept in the men's house, while women and children resided in family dwellings [75].

Household wealth differences can also have a substantial impact on the dwelling size [62, 100]. Unfortunately, variables describing this factor are missing in ethnographic databases. Although some proxies such as "Social Stratification [SCCS158, SCCS1751]" or "Number of Rich People [SCCS1721]" are available in *D-PLACE*, for using household wealth as control variable in AHFA-PMR analysis, more relevant data based on the deeper review of ethnographic literature are necessary.

The traditional residence typologies [5] are also problematic and have been criticized [16, 85, 101]. Many studies, including ours, focus simply on the most frequent or "ideal" residence type of a population in question and ignore intra-community variation. This is useful in cross-cultural comparisons but can be misleading when reconstructing actual residence patterns of prehistoric societies. First, there are often considerable differences between residence rules and actual practices within a community [16, 102]. Secondly, a couple often changes residence during their marriage, especially after one or more children are born (resulting in temporary matrilocal residence). Taking primary and alternative residence in later years together with residence in the first years of the marriage into account, Marlowe [4] concluded that majority of foragers (74%), as well as non-foragers (61%), were multilocal in the strict sense. Thirdly, residential rules apply differently to different community members. For example, in many matrilocal societies in Amazonia, chiefs and their sons usually resided patrilocally [103], and thus lived with more close kin than non-headmen [101]. Similarly, among Garo living in northeastern India, multiple residence patterns were present, which were all vital to Garo social structure. As Burling [94, pp. 215–216] puts it: "Some men must move in with their wives' families, while others must set up new households. Some men must move to their wives' villages, while others must bring their wives to their own villages. [. . .] Since it is not possible to say that any particular residence pattern is 'preferred,' it is unreasonable to demand that their custom be summed up by any such simple term as 'matrilocal'."

Lastly, with all cross-cultural studies based on ethnographic data, one needs to keep in mind that only a few studied societies were completely unaffected by colonialism or contact with modern civilization at the time of their description [104]. Most societies were exposed to various forms of cultural contact (e.g. epidemic diseases, the presence of missionaries, or trade with Westerners). These might have caused pacification, depopulation, changes in subsistence strategies or changes to social structure, including post-marital residence patterns or house size. It has been previously suggested that emergence of neolocality might have been caused by commercial exchange and industrialization [56], while ambilocality is often a result of depopulation [57]. On the other hand, neither prehistoric nor historical societies lived in complete isolation. Imported artefacts were common in almost every archaeological culture and recent evidence for plague in the Bronze Age in Eurasia [105] indicates that serious depopulations were not uncommon in pre-state societies.

## Conclusion

Our analysis confirms the cross-cultural association between house size and post-marital residence. Societies with larger dwellings tend to be matrilocal (compared to societies with smaller dwellings tending towards patrilocality). This association applies to broad range of post-marital residence patterns (not only to strictly matrilocal or patrilocal residence) and remains significant after controlling for other explanatory variables (agriculture, fixity of settlement, and construction material) and phylogeny. The effect of agriculture on dwelling size seems to be a by-product of the effect of fixity of settlement.

Further research is needed to evaluate the effect of other factors on house size, such as differences in household wealth, sociopolitical organization, functional differences in dwelling use, or western influence. Future research could also focus on distinction between residence in the husband's or the wife's parents' dwelling (patrilocal and matrilocal) and residence within the husband's or the wife's community (virilocal and uxorilocal). Comparing the dwelling size with other measures of residence, such as Helm's measure (i.e. the relative number of co-residing primary kin living with men versus women; [106]), could provide additional insight.

Our results suggest that average house floor area can be used as a material proxy for infer-ring post-marital residence patterns in prehistoric societies. That said, we agree with previous suggestions that "floor area alone should probably never be used as the sole index of residence" [45, p. 114] and that the correlations found "should only be used as working hypotheses to be tested with other lines of data" [47, p. 420]. Such data can be acquired using bioarchaeological methods (e.g. strontium and oxygen isotope or ancient DNA analyses) whose application in archaeological research has grown exponentially in recent years. Still, isotopic evidence must be interpreted with caution. Isotope analyses can distinguish mobility between different geo-logical regions, but not within one community or between communities living in regions with similar isotopic signal [18]. Interpreting isotope results in the terms of post-marital mobility is not always straightforward, since other types of mobility could lead to the same signal [107]. The evidence from cross-cultural and bioarchaeological analyses can complement each other, providing a more elaborated interpretation of the past social reality.

## Supporting information

**S1 File. List of sample societies and changes to the original variables.**
(DOCX)

**S1 Table. Model comparison for the evolution of AHFA and PMR based on Pagel's test for correlated evolution.**
(XLSX)

## Acknowledgments

We thank two anonymous reviewers for their comments and suggestions on improving the manuscript and to Alexander Barton for proofreading the text.

## Author Contributions

**Conceptualization:** Václav Hrnčíř, Pavel Duda.

**Data curation:** Václav Hrnčíř, Pavel Duda.

**Formal analysis:** Pavel Duda, Gabriel Šaffa.

**Funding acquisition:** Petr Květina, Jan Zrzavý.

**Investigation:** Václav Hrnčíř, Pavel Duda.

**Methodology:** Pavel Duda, Gabriel Šaffa, Jan Zrzavý.

**Project administration:** Pavel Duda, Jan Zrzavý.

**Supervision:** Petr Květina, Jan Zrzavý.

**Validation:** Pavel Duda.

**Visualization:** Václav Hrnčíř, Pavel Duda.

**Writing – original draft:** Václav Hrnčíř, Pavel Duda.

**Writing – review & editing:** Václav Hrnčíř, Pavel Duda, Gabriel Šaffa, Petr Květina, Jan Zrzavý.

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
