## [Decision Letter · Decision Letter 0]

10 Dec 2019

PONE-D-19-29673

Identifying post-marital residence patterns in prehistory: A phylogenetic comparative analysis of dwelling size

PLOS ONE

Dear Mr. Hrncir,

Thank you for submitting your manuscript to PLOS ONE. After careful consideration, we feel that it has merit but does not fully meet PLOS ONE’s publication criteria as it currently stands. Therefore, we invite you to submit a revised version of the manuscript that addresses the points raised during the review process.

all comments have to be addressed before re-submission.

We would appreciate receiving your revised manuscript by Jan 24 2020 11:59PM. To enhance the reproducibility of your results, we recommend that if applicable you deposit your laboratory protocols in protocols.io, where a protocol can be assigned its own identifier (DOI) such that it can be cited independently in the future. For instructions see: http://journals.plos.org/plosone/s/submission-guidelines#loc-laboratory-protocols

We look forward to receiving your revised manuscript.

Kind regards,

Peter F. Biehl, PhD

Academic Editor

PLOS ONE

Journal Requirements:

**When submitting your revision, we need you to address these additional requirements**:

**Please ensure that your manuscript meets PLOS ONE's style requirements, including those for file naming**. **The PLOS ONE style templates can be found at http://www.plosone.org/attachments/PLOSOne_formatting_sample_main_body.pdf and http://www.plosone.org/attachments/PLOSOne_formatting_sample_title_authors_affiliations.pdf**

Additional Editor Comments (if provided):

Your manuscript has now been seen by two referees, whose comments are appended below. You will see from these comments that while the referees find your work of potential interest, they have raised substantial concerns that must be addressed. In light of these comments, we cannot accept the manuscript for publication, but would be interested in considering a revised version that addresses these serious concerns.

We hope you will find the referees' comments useful as you decide how to proceed. Should presentation of further data and analysis allow you to address these criticisms, we would be happy to look at a substantially revised manuscript. However, please bear in mind that we will be reluctant to approach the referees again in the absence of major revisions.

Reviewers' comments:

Reviewer's Responses to Questions

**Comments to the Author**

1. Is the manuscript technically sound, and do the data support the conclusions?

Reviewer #1: Yes

Reviewer #2: Yes

2. Has the statistical analysis been performed appropriately and rigorously? 

Reviewer #1: Yes

Reviewer #2: Yes

3. Have the authors made all data underlying the findings in their manuscript fully available?

Reviewer #1: Yes

Reviewer #2: Yes

4. Is the manuscript presented in an intelligible fashion and written in standard English?

Reviewer #1: Yes

Reviewer #2: Yes

5. Review Comments to the Author

Reviewer #1: This is a very thorough, well written, clear, methodologically sound investigation of correlations between post-marital residence patterns and other aspects of social life, including stability of settlement and dwelling size. To judge the value of the paper there are some aspects do differentiate. One aspect is to investigate the argument, often used in anthropology and archaeology, that there is a systematic, or even causal relationship between matrilocality vs patrilocality and house sizes. What the authors do well is to improve the parameters of analyses, and to be more critical with the results, showing that there are more factors at play between the two targeted. What is not very clear is how that result helps prehistoric archaeology, beyond the fact that we need to be more cautious towards cross-cultural generalisations. That is surely a good point, but then the authors leave many problems inherent to the statistical, cross-cultural approach unexplored.

To judge the merits of the papers, I will argue on two levels. Level one is that I think the whole Idea of using this cross-cultural approach for archaeology is misleading. It makes, to me, absolutely no sense to take the results of this study and say, for example, the Early Neolithic LBK in Central Europe had large houses, thus we are to assume with a high probability that they had matrilocal marriage patterns. Even apart from the fact that – as the authors rightly state, Stable Isotope studies indicate the opposite, this is just not a good way of using anthropological models. However, I acknowledge that there are different views upon the merits of such a kind of approach, so I will review the paper putting aside my general disagreement.

If we look at what the authors are doing, the contribution is a valuable addition to the anthropological question about general relationships between postmarital residence patterns and house sizes. Especially, to move away from the binary opposition between matrilocality vs. patrilocality makes sense, including a wider array of models. Especially the control for dependencies between the case studies included is a step forward, and the phylogenetic approach to study the data points out some important caveats. It is important that they point out that the more important factor for house size is the stability of settlement. Also, using the phylogenetic approach, their observations about higher or lower stability of patterns (e.g. small houses and patrilocality is a more stable combination over time than others), helps clarify the significance of their findings. The explanations for these patterns do concern me, however. Arguments about general relations between married sisters, or unrelated brothers in law and how they would or would not thrive in larger or smaller houses seem very simplistic, at least it will not help win over sceptics (like me). As the authors are stating this with reference to the literature, maybe a little more nuanced, critical discussion would help. The main problem I have with the approach is the fact that they take two or three factors (house size, residence patterns and stability of settlement), and then discuss the question of causality between these. To me it seems that it would be very easy to miss other factors which could stand behind the correlations, but which are naturally excluded from the analyses because of the overall approach taken. The authors do discuss possible biases in the data. So a suggestion would be to put a little more thoughts into caveats. Could the oberservers bias be even more problematic? For example, the idea that ‘one society’ has basically one residence pattern could be a problematic assumption, maybe imposed by wertern observers? What other socio-economic factors could result in the correlation observed. What about the influence of colonialism? I do not recommend to change the argument, but in order to convince people like me, who value historical context more than cross-cultural statistical patterns, such a more exhaustive discussion would surely benefit the paper.

Apart from that I have only minor comments:

Page 2 of the main text, after mentioning Eulau and the Lech River examples, also Sjögren et al. 2016 Diet and Mobility in the Corded Ware of Central Europe, PLOS ONE

On the same page, after references 27-30: Maybe mention and briefly discuss the issue of the world-wide phenomenon of reduction of male haplogroups (y-haplogroups), discussed in Zeng et al., Cultural hitchhiking and competition between patrilineal kin groups explain the post-Neolithic Y-chromosome bottleneck, Nature Comm. 2018

Page 3: The Greek Neolithic does not start at 6800, but rather 6600/6500 BC

Page 4: The section Methods is immediately followed by the subheader Data description. This sounds illogical. Also, what follows is not acutally a description of the data, but a discussion about how to best represent the data, how to use data as proxy. That makes totally sense within the method chapter, so best change the subheader into something that makes it clear we deal with proxy definition, or data representation, which makes more sense in the Methods chapter than description.

In the Conclusion the authors very briefly hints at the new possibility of actually studying prehistorc residence patterns and their relation to house sizes, through Isotopes and aDNA. This is a good point, and maybe worth expanding more. Because this really brings home the relevance of the paper apart from the question whether the statistical inferences made have any relevance for individual archaeological case studies.

So, overall, putting aside my own ideological differences with the approach, I would think that within the framework set, this paper is well written and methodologically sound, and I recommend accept with minor revisions.

Reviewer #2: This paper presents a cross-cultural study of dwelling size from the archaeological and ethnographic records to make inferences about post-marital residence (PMR) patterns. The study heavily builds on previous research, and presents additional data and a phylogenetic comparative analysis as original contributions.

Overall, the paper is clearly organized, with some redundancies particularly between the Introduction, Discussion, and Conclusion sections. The Introduction is an excellent overview of the past and recent research on PMR. However, limitations of PMR studies, especially as they relate to the archaeological record, should be summarized here, then discussed in relation to the results of the current study in the Discussion sections.

This latter structural change would be important for several reasons, reflecting also some shortcomings of the manuscript. Most importantly, as noted by the authors briefly on Page 2 (“…changes in AHFA in response to changes in PMR or other aspects of social organization”) and discussed to some extent on Pages 8 and 10, PMR is only one of the potential variables that could impact dwelling size. Sufficient to say, in recent, cross-cultural studies dwelling size is used as a proxy for household wealth, and thus, the number of household members does not definitely positively correlate with the physical extent of the dwellings (see e.g., Kohler et al. 2017: doi: 10.1038/nature24646; Kohler and Smith (eds.) 2018: https://uapress.arizona.edu/book/ten-thousand-years-of-inequality). For other perspectives, including functional differences in residential use as they relate to dwelling size and organization, see Coupland and Banning (eds.) 1996 (https://www.amazon.com/People-Who-Lived-Houses-Archaeological/dp/1881094154) and Burke (ed.) 2016 (https://journals.openedition.org/palethnologie/476). In addition, as the authors also note in the Discussion section, the use of average house floor area derived from regional scale datasets, being archaeological or ethnographic, necessarily eradicates variability at the local scale as well as ignores the high degree of heterogeneity in cultural practices. These aspects might largely account for some inconclusive patterns in the figures and the ambiguity of the interpretation of the statistical data.

Finally, the authors compare societies with fundamentally different sociopolitical settings, from hunting-gatherer bands to state-level societies. Although I clearly understand that this is out of the scope of the current manuscript, sociopolitical organization could be considered as an analytical variable to improve PMR studies in the future.

To sum up, this paper is an important contribution to PMR studies and, by considering the proposed additions and changes, it has the potential to serve as a starting point to study dwelling size as a proxy for cultural, social, and economic settings from a more integrative point of view. Therefore, I recommend the manuscript for publication with minor changes.

Minor issues:

Throughout the paper: `nomadic` frequently is mistakenly used instead of `mobile` or `non-sedentary`

Throughout the paper: consider using `permanency of settlement` instead of `stability of settlement`

Page 2: “In this study, we use “patrilocality” for both patri- and virilocality and “matrilocality” for both matri- and uxorilocality, since they cannot be distinguished in prehistoric patterns by the currently available methods.” Since the study focuses on residence in the husband’s father’s or the wife’s kin’s dwellings, virilocality and uxorilocality must be out of scope.

Table 1, Page 5, Settlement column: the ‘Original scale’ classification scheme and terms must be reconsidered, especially 1-4 and 5-6.

Figure 3: reconsider color coding. Based on Figures 1 and 2 (as indicated in the caption), the interpretation is difficult.

6. PLOS authors have the option to publish the peer review history of their article (what does this mean?). If published, this will include your full peer review and any attached files.

Reviewer #1: No

Reviewer #2: No

---

## [Author Response · Author response to Decision Letter 0]

13 Jan 2020

We thank the reviewers for their comments and suggestions on the manuscript. We have addressed each of these comments below and in the revised manuscript. We have highlighted (using gray highlight) all sections of the manuscript that have been revised.

Reviewer #1: One aspect is to investigate the argument, often used in anthropology and archaeology, that there is a systematic, or even causal relationship between matrilocality vs patrilocality and house sizes. What the authors do well is to improve the parameters of analyses, and to be more critical with the results, showing that there are more factors at play between the two targeted. What is not very clear is how that result helps prehistoric archaeology, beyond the fact that we need to be more cautious towards cross-cultural generalisations. That is surely a good point, but then the authors leave many problems inherent to the statistical, cross-cultural approach unexplored. To judge the merits of the papers, I will argue on two levels. Level one is that I think the whole Idea of using this cross-cultural approach for archaeology is misleading. It makes, to me, absolutely no sense to take the results of this study and say, for example, the Early Neolithic LBK in Central Europe had large houses, thus we are to assume with a high probability that they had matrilocal marriage patterns. Even apart from the fact that – as the authors rightly state, Stable Isotope studies indicate the opposite, this is just not a good way of using anthropological models. However, I acknowledge that there are different views upon the merits of such a kind of approach, so I will review the paper putting aside my general disagreement.

- We do not share Reviewer 1 skepticism towards applicability of cross-cultural approach in archaeology. In our study we see the potential to draw attention to the complexity of kinship systems (with PMR being a part of it). There is not a single way in which archaic societies infer family ties. We see potentially dangerous efforts to view bioarchaeological data (e.g., Sr isotopes) as direct and only indicators of the PMR and kinship system (see for example Furholt 2017; https://doi.org/10.1515/pz-2017-0024). If the cross-cultural trend shows different result than the bioarchaeological analysis, this does not necessarily mean one of the approaches is misleading. Conversely, this may lead to a more elaborate interpretation of the past social reality (e.g. Early Neolithic society in Central Europe).

Reviewer #1: The explanations for these patterns do concern me, however. Arguments about general relations between married sisters, or unrelated brothers in law and how they would or would not thrive in larger or smaller houses seem very simplistic, at least it will not help win over sceptics (like me). As the authors are stating this with reference to the literature, maybe a little more nuanced, critical discussion would help. 

- The arguments have been expanded and discussed more critically (p. 9-10).

Reviewer #1: The main problem I have with the approach is the fact that they take two or three factors (house size, residence patterns and stability of settlement), and then discuss the question of causality between these. To me it seems that it would be very easy to miss other factors which could stand behind the correlations, but which are naturally excluded from the analyses because of the overall approach taken. The authors do discuss possible biases in the data. So a suggestion would be to put a little more thoughts into caveats. Could the oberservers bias be even more problematic? For example, the idea that ‘one society’ has basically one residence pattern could be a problematic assumption, maybe imposed by wertern observers? What other socio-economic factors could result in the correlation observed. What about the influence of colonialism? I do not recommend to change the argument, but in order to convince people like me, who value historical context more than cross-cultural statistical patterns, such a more exhaustive discussion would surely benefit the paper.

- The issues of observers bias (one society = one residence pattern) and similarity of cultures due to colonialism have been mentioned in the discussion (p. 11-12.) That said, the role of similarity of societies due to contact with western cultures might be exaggerated. A recent cross-cultural study, which sought to explain variance in daily foodsharing norms across societies (Ringen et al. 2019, https://doi.org/10.1016/j.evolhumbehav.2019.04.003) has controlled for both the phylogeny and non-independence due to the time at which data were collected (the ‘ethnographic present’), assuming that the longer the societies were exposed to western influence, the more similar they became to each other. While the phylogeny explained moderate amount of variance in food sharing norms, ethnographic present did not, suggesting that this is not the case, at least for some cultural characteristics of the societies.

Reviewer #2: The Introduction is an excellent overview of the past and recent research on PMR. However, limitations of PMR studies, especially as they relate to the archaeological record, should be summarized here, then discussed in relation to the results of the current study in the Discussion sections.

- We thank the reviewer for this assessment of the Introduction section. The main goal of our study is to investigate the AHFA-PMR correlation found by previous cross-cultural studies. That is why we describe methodological limitations of this type of studies in the Introduction. We feel that the flow of the ideas throughout the paper is better when we discuss other limitations of PMR studies in light of our results in the discussion section. This way we avoid unnecessary repetition from having all the limitations described in the Introduction.

Reviewer #2: This latter structural change would be important for several reasons, reflecting also some shortcomings of the manuscript. Most importantly, as noted by the authors briefly on Page 2 (“…changes in AHFA in response to changes in PMR or other aspects of social organization”) and discussed to some extent on Pages 8 and 10, PMR is only one of the potential variables that could impact dwelling size. Sufficient to say, in recent, cross-cultural studies dwelling size is used as a proxy for household wealth, and thus, the number of household members does not definitely positively correlate with the physical extent of the dwellings (see e.g., Kohler et al. 2017: doi: 10.1038/nature24646; Kohler and Smith (eds.) 2018: https://uapress.arizona.edu/book/ten-thousand-years-of-inequality). For other perspectives, including functional differences in residential use as they relate to dwelling size and organization, see Coupland and Banning (eds.) 1996 (https://www.amazon.com/People-Who-Lived-Houses-Archaeological/dp/1881094154) and Burke (ed.) 2016 (https://journals.openedition.org/palethnologie/476). In addition, as the authors also note in the Discussion section, the use of average house floor area derived from regional scale datasets, being archaeological or ethnographic, necessarily eradicates variability at the local scale as well as ignores the high degree of heterogeneity in cultural practices. These aspects might largely account for some inconclusive patterns in the figures and the ambiguity of the interpretation of the statistical data.

Reviewer #2: Finally, the authors compare societies with fundamentally different sociopolitical settings, from hunting-gatherer bands to state-level societies. Although I clearly understand that this is out of the scope of the current manuscript, sociopolitical organization could be considered as an analytical variable to improve PMR studies in the future.

- We have incorporated these suggestions and additional references into the Introduction and (primarily) the Discussion section, while keeping the general structure of the manuscript intact.

There is no doubt a strong link between AHFA or generally household size and household wealth. However, this relationship manifests itself mainly in intra-societal level, where there is a difference between larger houses of the rich and smaller houses of the poor. We have studied societies where large houses are a standard part of material culture. Such societies were then compared among themselves. The variability of the size of houses within societies was beyond the focus of our study.

Reviewer #1: Page 2 of the main text, after mentioning Eulau and the Lech River examples, also Sjögren et al. 2016 Diet and Mobility in the Corded Ware of Central Europe, PLOS ONE

- The reference has been added (p. 2).

Reviewer #1: On the same page, after references 27-30: Maybe mention and briefly discuss the issue of the world-wide phenomenon of reduction of male haplogroups (y-haplogroups), discussed in Zeng et al., Cultural hitchhiking and competition between patrilineal kin groups explain the post-Neolithic Y-chromosome bottleneck, Nature Comm. 2018

- This issue has been mentioned in the Introduction section (p. 2).

Reviewer #1: Page 3: The Greek Neolithic does not start at 6800, but rather 6600/6500 BC

- The dating has been changed accordingly (p. 3).

Reviewer #1: Page 4: The section Methods is immediately followed by the subheader Data description. This sounds illogical. Also, what follows is not acutally a description of the data, but a discussion about how to best represent the data, how to use data as proxy. That makes totally sense within the method chapter, so best change the subheader into something that makes it clear we deal with proxy definition, or data representation, which makes more sense in the Methods chapter than description.

- The subheader has been changed (p. 4).

Reviewer #1: In the Conclusion the authors very briefly hints at the new possibility of actually studying prehistorc residence patterns and their relation to house sizes, through Isotopes and aDNA. This is a good point, and maybe worth expanding more. Because this really brings home the relevance of the paper apart from the question whether the statistical inferences made have any relevance for individual archaeological case studies.

- The application of isotopes and DNA for studying prehistoric residence patterns has been mentioned in the Introduction section (p. 2). We have expanded the Conclusion section, where we touch upon some limitations of these methods (especially isotopes), arguing for multidisciplinary approach (p. 12).

Reviewer #2: Throughout the paper: `nomadic` frequently is mistakenly used instead of `mobile` or `non-sedentary`

- The term `nomadic` has been changed to `mobile` throughout the paper.

Reviewer #2: Throughout the paper: consider using `permanency of settlement` instead of `stability of settlement`

- Throughout the paper, we use `permanency` in relation to building material. To avoid confusion, we changed `stability of settlement` to `fixity of settlement`.

Reviewer #2: Page 2: “In this study, we use “patrilocality” for both patri- and virilocality and “matrilocality” for both matri- and uxorilocality, since they cannot be distinguished in prehistoric patterns by the currently available methods.” Since the study focuses on residence in the husband’s father’s or the wife’s kin’s dwellings, virilocality and uxorilocality must be out of scope.

- It is true that the two suggested explanations relate specifically to patrilocality and matrilocality (residence in the husband’s father’s or the wife’s kin’s dwellings). On the other hand, we did not limit our analyses only to patrilocal or matrilocal societies. We also included virilocal and uxorilocal societies, among others, to measure the tendency towards matrilocality. Besides, PMR is not properly distinguished in original data source (D-PLACE). Although both states (patrilocality and virilocality) are present, they have somewhat different definitions here than in the paper. In the expanded conclusion section, we suggest that future research could focus on finer distinction between patrilocal and virilocal societies.

Reviewer #2: Table 1, Page 5, Settlement column: the ‘Original scale’ classification scheme and terms must be reconsidered, especially 1-4 and 5-6.

- Column ‘Original scale’ describes trait states as described in the original source (D-PLACE). Therefore, this classification scheme and terms should not be changed.

Reviewer #2: Figure 3: reconsider color coding. Based on Figures 1 and 2 (as indicated in the caption), the interpretation is difficult.

- We believe that the color coding in Fig 3A which correspond to Fig 1 and reflects PMR states is appropriately chosen. In Fig 2, the colors reflect reconstructed ancestral states. We agree that this is confusing and have changed the caption accordingly (p. 7). In Fig. 3B, an explanatory variable is settlement, which is not included in Fig. 1. Therefore, no color coding has been assigned. We believe that adding color coding to Fig 3B would confuse the audience, rather than facilitate interpretation of the figure.

---

## [Decision Letter · Decision Letter 1]

5 Feb 2020

Identifying post-marital residence patterns in prehistory: A phylogenetic comparative analysis of dwelling size

PONE-D-19-29673R1

Dear Dr. Hrncir,

We are pleased to inform you that your manuscript has been judged scientifically suitable for publication and will be formally accepted for publication once it complies with all outstanding technical requirements.

With kind regards,

Peter F. Biehl, PhD

Academic Editor

PLOS ONE

Additional Editor Comments (optional):

Reviewers' comments:

Reviewer's Responses to Questions

**Comments to the Author**

1. If the authors have adequately addressed your comments raised in a previous round of review and you feel that this manuscript is now acceptable for publication, you may indicate that here to bypass the “Comments to the Author” section, enter your conflict of interest statement in the “Confidential to Editor” section, and submit your "Accept" recommendation.

Reviewer #1: All comments have been addressed

Reviewer #2: All comments have been addressed

2. Is the manuscript technically sound, and do the data support the conclusions?

Reviewer #1: Yes

Reviewer #2: Yes

3. Has the statistical analysis been performed appropriately and rigorously? 

Reviewer #1: I Don't Know

Reviewer #2: Yes

4. Have the authors made all data underlying the findings in their manuscript fully available?

Reviewer #1: Yes

Reviewer #2: Yes

5. Is the manuscript presented in an intelligible fashion and written in standard English?

Reviewer #1: Yes

Reviewer #2: Yes

6. Review Comments to the Author

Reviewer #1: (No Response)

Reviewer #2: The comments have been addressed and the manuscript has been improved.

I suggest an additional, minor change: in Table 1 'nomadic' and `seminomadic' should be changed to 'mobile' and 'semi-sedentary', respectively.

Other than that, I recommend the manuscript for publication without further modifications.

7. PLOS authors have the option to publish the peer review history of their article (what does this mean?). If published, this will include your full peer review and any attached files.

Reviewer #1: Yes: Martin Furholt

Reviewer #2: No

---

## [Editor Report · Acceptance letter]

11 Feb 2020

PONE-D-19-29673R1 

Identifying post-marital residence patterns in prehistory: A phylogenetic comparative analysis of dwelling size 

Dear Dr. Hrncir:

I am pleased to inform you that your manuscript has been deemed suitable for publication in PLOS ONE. Congratulations! Your manuscript is now with our production department. 

With kind regards,

on behalf of

Dr. Peter F. Biehl 

Academic Editor

PLOS ONE